# Peripheral Regional Anesthesia Using Local Anesthetics: Old Wine in New Bottles?

**DOI:** 10.3390/jcm12041541

**Published:** 2023-02-15

**Authors:** Lukas Gasteiger, Lukas Kirchmair, Elisabeth Hoerner, Ottokar Stundner, Markus W. Hollmann

**Affiliations:** 1Department of Anesthesia and Critical Care Medicine, Medical University of Innsbruck, 6020 Innsbruck, Austria; 2Department of Anesthesia and Critical Care Medicine, Hospital Schwaz, 6130 Schwaz, Austria; 3Department of Anesthesiology, Amsterdam University Medical Center, University of Amsterdam, 1100 Amsterdam, The Netherlands

**Keywords:** regional anesthesia, local anesthetics, adjuvants, prolongation of action

## Abstract

During the past decade, numerous efforts were undertaken aiming at prolonging the analgesic effect of regional anesthesia. With the development of extended-release formulations and enhanced selectivity for nociceptive sensory neurons, a very promising contribution to the development of pain medications has been achieved. At present, liposomal bupivacaine is the most popular, non-opioid, controlled drug delivery system, but its duration of action, which is still controversially discussed, and its expensiveness have decreased initial enthusiasm. Continuous techniques can be seen as an elegant alternative for providing a prolonged duration of analgesia, but for logistic or anatomical reasons, they are not always the best choice. Therefore, focus has been directed towards the perineural and/or intravenous addition of old and established substances. As for perineural application, most of these so-called ‘adjuvants’ are used outside their indication, and their pharmacological efficacy is often not or only poorly understood. This review aims to summarize the recent developments for prolonging the duration of regional anesthesia. It will also discuss the potential harmful interactions and side effects of frequently used analgesic mixtures.

## 1. Introduction

With more than 80% of patients experiencing acute pain following surgical procedures, and half of them reporting inadequate pain therapy, the control and management of post-surgical pain remains one of the most challenging aims of modern anesthesia [1]. Post-surgical pain and immobilization reinforce each other and lead to low patient satisfaction, delayed recovery and discharge from hospital. Thus, both reflect main targets for Enhanced Recovery after Surgery (ERAS) pathways [2,3,4]. For decades, opioids, with their potent analgesic effectiveness, have been the mainstay of postoperative pain management. However, opioids may also lead to adverse events, e.g., prolonged length of hospital stay (LOS) due to dose-related side effects such as respiratory depression, sedation, postoperative nausea and vomiting (PONV), tolerance and hyperalgesia, urinary retention or the development of bowel dysfunction [5,6,7]. Additionally, when considering the current opioid crisis, effective pain control considerably reduces the need for opioids after surgical procedures.

Therefore, modern concepts of multimodal pain therapy, including minimal-invasive approaches in surgery, regional anesthesia (RA) and non-opioid pain medication, aim at facilitating different pathways in order to provide effective pain control, early mobilization and minimize opioid demand, leading to faster patient recovery following major surgery.

RA techniques have shown their benefit by reducing opioid use and shortening the stay on postoperative care units [2,8]. However, a single injection of local anesthetics (LA) hardly lasts longer than 24 h. A common alternative is the use of catheter-based anesthesia techniques, for which prolonged analgesia and shorter LOS have been described [9]. Unfortunately, these techniques have some disadvantages, amongst others: catheter tip dislocation, infection and the need for more complex in-hospital logistics [2].

Several pharmacological and application strategies have therefore been assessed, with the primary goal to achieve a safe, reliable and long-lasting analgesic effect with minimal motoric restrictions, enabling the patient to be as autonomous as possible. Promising solutions incorporate LAs and/or analgesics into biodegradable structures that provide extended release of the drug at the target location with low systemic side effects. In addition, progress has been made in the field of selectively blocking sodium channels, which play a central role in the genesis and conduction of nociceptive stimuli.

Here, we review the current literature to outline the most promising strategies and discuss their benefits and limitations.

## 2. Strategies

### 2.1. Innovations in Pharmacology

The easiest way to provide an ‘ideal’ RA would be the use of an LA with the following properties: an analgesic effect that lasts at least as long as the acute postoperative pain (>24 h), minimal systemic side effects and a selectivity to sensory properties that does not lead to the motoric and autonomous impairment of the patient. We do not have such a substance yet, but several efforts have led to achievements that have already come quite close.

#### 2.1.1. Conventional LAs

Most LAs realize their effect by binding inside neuronal, voltage-dependent sodium channels (Na_v_) consisting of nine subtypes (Na_v_ 1.1–1.9) that lead to nerve conduction blockades by impeding cell depolarization caused by Na^+^ influx inhibition [10,11].

The effect of LAs is defined by three specific properties of each substance. Pharmacokinetic properties such as speed of onset and duration of action are defined by physiochemical characteristics; a p*K_a_* close to physiological extracellular pH leads to stronger permeability. The lipophilicity of an LA is decisive for its potency or analgesic effect. Finally, protein binding is also responsible for the duration of action as it prevents the metabolization of free LA [12].

Ester-type LAs such as procaine, chloroprocaine or tetracaine, for which the binding between the hydrophilic amino group and the lipophilic benzene ring is constituted by an ester, generally have a short to moderate duration of action. For amid-type LAs such as lidocaine, prilocaine, ropivacaine and bupivacaine, the link is constituted by an amid group and characterized by a moderate to long duration of action, with ropivacaine and bupivacaine lasting up to almost 12 h [11,13,14].

LAs can easily diffuse into the central nervous system (CNS) through the blood–brain barrier and interact not exclusively with peripheral nerves. Therefore, highly perfused organs such as the heart or brain can be severely affected if larger plasma levels of LAs are built up due to overdosing or reduced protein binding following accidental intravascular injection [15,16,17]. Typical CNS symptoms of LA systemic toxicity (LAST) are perioral paresthesia, dizziness, slurred speech and metallic taste. Also, severe CNS symptoms such as seizures or loss of consciousness have been reported [12]. Cardiovascular toxicity is caused by a reduction of action potential duration and the refractory period, leading to negative inotropic and dromotropic effects in the heart [18,19,20]. LAST is the limiting factor for the dosing of LAs.

#### 2.1.2. Extended-Release Formulations

Extended-release formulations were developed to allow for larger doses of LAs to be released slowly over a longer time period, thereby prolonging analgesic effects and avoiding systemic side effects [21]. LAs can be encapsulated in implantable drug delivery systems that facilitate a constant release and minimal tissue reaction, and they should be biodegradable as non-toxic and excretable products [13]. An overview of the approved extended-release formulations is given in Table 1.

##### Liposomal Formulation of Extended Release

Liposomes are phospholipid-based nanovesicles and have already been used for drug delivery in chemotherapy and in the treatment of different infectious diseases. They consist of a bilayer “sealed sack” that can contain hydrophilic substances inside or lipophilic substances within the bilayer [12,13,33]. Liposomes can be classified as unilamellar, multilamellar or multivesicular [13]. Multivesicular liposomes are non-concentric liposomes with several vesicles closely packed in the outer layer similar to a honeycomb, and they are produced with DepoFoam^®^ technology and encapsulate the pharmaceutical ingredient [34]. This structure leads to higher stability, allows for controlled slow-release properties and avoids a burst release of the active drug [34,35].

Liposomal bupivacaine (EXPAREL^®^, Pacira Pharmaceuticals, Parsippany, NJ, USA) was the first extended-release liposomal LA approved by the American Food and Drug Administration (FDA). EXPAREL^®^ is a multivesicular liposome that has been approved for wound infiltration (2011) and for interscalene brachial plexus block (2018), and it is described to provide an analgesia duration of up to 72 h [12,22,23]. It has also been approved by the European Medicines Agency (EMA) for field block infiltration for both femoral and brachial plexus blocks in 2020 [36]. With these properties, EXPAREL^®^ was expected to become an ideal tool for controlled, extended drug delivery. However, recent trials and meta-analyses found a prolongation of analgesia that was statistically significant but was deemed to have no clinically relevant extended duration of action compared to plain bupivacaine [22,23,36,37,38,39,40,41,42,43,44,45,46,47,48,49].

##### Biological Polymers

Bupivacaine/Meloxicam (ZYNRELEF^®^; HERON Therapeutics, San Diego, CA, USA) is a synergistic fixed-dose combination of bupivacaine and the non-steroidal anti-inflammatory drug (NSAID) meloxicam that is incorporated into biodegradable polymers and was approved by the EMA in 2020 and by the FDA in 2021 for needle-free application at the surgical site [25,26,27]. Results from recent randomized phase III trials (EPOCHE I and EPOCHEII) indicate improved postoperative pain control and a reduced need for opioids, resulting in less opioid-related adverse events [50,51,52]. However, it should be noted that the primary endpoint was the mean area under the curve (AUC) of the numerous rating scale (NRS), with opioid consumption being just a secondary endpoint in both studies. Given the restricted number of randomized clinical trials, more data are needed to better evaluate its properties and efficacy.Sucrose acetate isobutyrate extended-release bupivacaine (POSIDUR^TM^, SABER^®^ Bupivacaine; DURECT, Cupertino, CA, USA) was approved by the FDA in 2021 for subacromial injection under direct arthroscopic guidance, following safety and efficacy determination in previous phase IIb and III trials [28,29,30]. Recently, a double-blinded randomized trial assessed pain intensity during 90° shoulder flexion with opioid consumption over 72 h in 78 patients undergoing arthroscopic subacromial decompression. The authors described a reduction in pain and opioid consumption, as well as prolonged time until rescue opioid analgesia for SABER^®^ Bupivacaine when compared to placebo and local bupivacaine hydrochloride (HCl) infiltration [53].In 2021, a bioresorbable collagen implant containing bupivacaine hydrochloride (INL-001; XARACOLL^®^, Innocoll Holdings Limited, Princeton, NJ, USA) received US FDA approval for placement into the surgical area during open inguinal hernia repair [31]. Results from two double-blinded randomized phase III studies (MATRIX I and MATRIX II) with 624 patients scheduled for unilateral hernia repair assessed the sum of the pain intensity during the first 24 h (SPI24). Both trials reported a lower SPI24 and lower opioid consumption over 24 h for INL-001 compared to placebo [32]. A recently published trial assessing pharmacokinetic properties of INL-001 described a prompt and continuous release of bupivacaine over 96 h [54].

#### 2.1.3. Sodium Channel Selectivity

Voltage-gated ion channels are transmembrane proteins that allow for ions to move along an electrochemical gradient across cellular membranes. Voltage-gated sodium (Na_v_) channels generate sodium currents that play an essential role for the initiation and transmission of action potentials among different excitable tissues and have various regulatory properties [11]. To date, nine Na_v_ 1 channels—Na_v_ 1.1 to 1.9—have been identified and exhibit tissue-specific expression profiles. Na_v_ 1.1, 1.2 and 1.3, for example, are expressed solely in the CNS, whereas Na_v_ 1.6 can be found in both the peripheral nerval system and CNS. Na_v_ 1.7, 1.8 and 1.9 are restricted to peripheral nerves, and Na_v_ 1.4 and 1.5 are expressed exclusively on skeletal and cardiac muscle cells [55,56,57]. At present, traditional LAs show little selectivity among all these Na_v_ channels, and their limiting factors are cardiac or central nervous toxicity [11].

Hence, special interest was focused on the subtypes Na_v_ 1.7, 1.8 and 1.9—especially Na_v_ 1.7—known to be involved in the transmission of nociception in peripheral nerves, with a high concentration in the dorsal root ganglion [58,59]. Mutations in genes that encode for Na_v_ channel subunits have been shown to change the functional properties of the channel, causing channel dysfunction. Specific mutations that lead to a loss of function in Na_v_ 1.7 channels are associated with a congenital indifference to pain, whereas those mutations causing a gain of function are being found in hereditary pain disorders [56,60]. Drawing upon these findings, a lot of attention has been focused on Na_v_ channels as potential targets for the further development of pain medications.

Interestingly, Na_v_ channels have ever since been molecular targets for numerous natural neurotoxins, such as tetrodotoxine (TTX), saxitoxin (STX), batrachotoxin (BTX) and other peptide toxins found in various poisonous animals [61]. The mode of action of TTX was first described in 1960 by Naharashi et al., who reported that TTX inhibits Na_v_ channels at very low concentrations [62]. The subsequent discrimination between TTX-sensitive (Nav 1.1–1.4, 1.6 and 1.7) and TTX-insensitive (1.5, 1.8, and 1.9) Na_v_ channels was a milestone that marked a new chapter in pain medicine [63].

At present, several selective inhibitors for Na_v_ 1.7 and Na_v_ 1.8 are being assessed, including pore blockers (TTX and STX), sulfonamides, peptides and monoclonal antibodies [11,64]. For the two natural toxins, TTX and STX, inhibition of pain has already been demonstrated in preclinical studies [58]. Their high sensitivity to Na_v_ 1.7 and low sensitivity to Na_v_ 1.5 opens an intriguing scope of application. Additionally, animal studies have demonstrated a long duration of action and synergistic effects when combined with traditional LAs [65]. In a small trial with 10 volunteers, the STX analogue Neosaxitoxin (NeoSTX) achieved a neural block for cold pain detection that lasted 24 h when subcutaneously injected [66]. In a subsequent study, the same study group described longer analgesia for preincisional infiltrated NeoSTX when compared to infiltrated bupivacaine after laparoscopic cholecystectomy [67]. A recent randomized double-blinded phase I trial that included 84 volunteers showed a prolonged neural blockade for a combination of NeoSTX and bupivacaine when compared to NeoSTX-saline, bupivacaine or placebo [68].

## 3. Clinical Concepts

### Mixture of LAs

The practice of mixing LAs to combine the characteristics of two substances has a long tradition and goes back to the 1950s [69]. The combination of short- and long-acting drugs should accelerate block onset to rapidly obtain surgical anesthesia on the one hand, and prolong block duration to achieve postoperative analgesia on the other hand. The two components are mixed in one syringe or injected sequentially. Both methods show similar results regarding block characteristics [70,71]. Basically, the driving force of an LA to penetrate neural tissue is its concentration gradient. The combination of two substances results in the dilution of each component, lowering its concentration and thus limiting its transfer across the cell membrane [72]. Moreover, mixing solutions of varying physicochemical properties (e.g., pH) results in changes to their ionized and non-ionized fractions [72,73]. LAs can cross lipid layers only in their non-ionized form. Thus, lowering the pH as a result of mixing two LAs (e.g., lidocaine and bupivacaine) decreases the efficacy of each or both components [72]. Some examples of investigated compounds in currently available LAs are as follows: lidocaine/bupivacaine, lidocaine/ropivacaine, mepivacaine/bupivacaine, mepivacaine/ropivacaine, chloroprocaine/bupivacaine, prilocaine/bupivacaine and prilocaine/ropivacaine. Such mixtures are used for several regional techniques, such as epidural and caudal anesthesia, as well as peripheral nerve blocks (PNB). Clinical and experimental studies from the last century obtained conflicting results. In common, they all showed an accelerated block-onset when a short-acting drug was added to a long-acting drug, but at the expense of block duration [69,74,75]. These findings have been reproduced in recent studies [76,77].

Since peripheral RA has changed significantly with the implementation of ultrasound-guided peripheral nerve block (US-PNB) techniques, block onset is of limited clinical relevance. US-PNB is known to reduce the time to onset of sensory block. In contrast to traditional nerve localization techniques, high-resolution ultrasound imaging has led to the precise injection and monitoring of LA spread around nerves [78,79]. Thus, even long-acting drugs show acceptable onset times. Accordingly, US-PNB allows for the reduction in dosage of LA and a reduced potential for direct local LA toxicity [80]. Thus, the initial idea of combining the advantages of two drugs (rapid block onset and long-lasting analgesia) cannot be supported in the light of modern regional anesthetic techniques. Furthermore, the systemic toxicity of two injected LAs is regarded to be additive [81]. This issue is often neglected in clinical practice. Additionally, mixing LAs might increase their neurotoxic potential [82].

## 4. Additives/Adjuvants

In order to prolong the duration of analgesia, the co-application of several additives that supposedly extend analgesia were studied [83]. Some additives are also used to shorten the time to the onset of LA, such as sodium bicarbonate, which is often added to short- or intermediate-acting LAs (e.g., lidocaine). As this review focusses on the prolongation of sensory block in PNB, additives used to shorten the time to onset will not be further discussed. LA additives are defined as single- or multiple-additive pharmacological agents, administered perineurally or systemically [84]. Different drugs are commonly used, including α_2_-receptor agonists, opioids, corticosteroids or vasoconstrictors. However, one should consider that mixing substances of different substance classes creates a new drug that lacks registration, changes their pharmacological properties and may also lead to precipitation [72,85]. In our mind, mixtures should only be created after the strict estimation of their safety profile. Therefore, in this section, we will only discuss the most common substances that are deemed to be at least non-hazardous and omit those with suspected neurotoxic effects (e.g., midazolam, magnesium) or unwanted side effects such as nausea and hallucinations (e.g., ketamine, neostigmine) (Table 2) [86].

Also, it should be kept in mind that by using drugs for other applications other than those covered by the approval of medical regulatory authorities such as the FDA or EMA, the national drug safety legislation should be respected. In some countries, special patient consent for off-label use is mandatory.

### 4.1. Imidazoline Derivates

#### 4.1.1. Clonidine

Clonidine is a frequently used and well-studied drug that has mainly non-selective α_2_ adrenoreceptor-agonistic properties (α_2_: α_1_ activity ratio 200:1). It is known to produce analgesic, hemodynamic and sedative effects [84]. Clonidine binds to α_2_-receptors in the dorsal horn of the spinal cord. It is also deemed to have effects on peripheral nerves by inhibiting the hyperpolarization-activated cation current (I_h_) by blocking nucleotide-gated channels. The I_h_ is involved in restoring a resting potential for subsequent action potentials in neurons following hyperpolarization [87].

Perineural vasoconstriction increases LA concentration by lowering blood flow, thereby slowing the reabsorption of LA. Due to its α_1_-adrenoreceptor agonistic properties, vasoconstriction may be a mechanism of action when clonidine is used as an additive. In a recent meta-analysis, the prolongation of RA ranged between 2.8 and 3.3 h for sensory blockades, motoric blockades and time until first request for additional pain medication [8].

However, there are conflicting results regarding the potential of clonidine to prolong nerve block duration. Out of 27 studies included in a qualitative review, 15 supported the perineural use of clonidine, whereas 12 did not find any benefit [88]. In addition, there have been conflicting results depending on the location of the PNB. Prolonged duration was described when clonidine was added to ropivacaine in a brachial plexus block, but this was not the case when it was added to levobupivacaine in a sciatic nerve block [89,90,91]. Common side effects include sedation, hypotension and bradycardia, which limit the use of clonidine for day-care surgery [92]. Therefore, doses should not exceed 0.5–1 mcg/kg of ideal body weight [88,92].

#### 4.1.2. Dexmedetomidine

Dexmedetomidine is also a non-selective α_2_-adrenoreceptor agonist with a much higher selectivity for the α_2_-receptor ((α_2_: α_1_ activity ratio 1620:1) compared to clonidine [93]. When perineurally administered, it exerts effects similar to clonidine; however, less vasoconstriction occurs due to lower α_1_ agonism. Different meta-analyses reported a prolonged duration of analgesia by almost five hours when combined with LAs [94,95]. Similar to clonidine, adverse effects such as bradycardia, hypotension, sedation and prolonged motoric blockade must be expected and monitored, which also limits its use in day-care surgery.

### 4.2. Dexamethasone

Dexamethasone is a strong glucocorticoid with anti-inflammatory activity. It is also known as an anti-emetic drug. During the last decade, it has been extensively studied as an additive for PNBs, showing a prolongation of analgesia of six to ten hours when combined with intermediate- and long-acting LAs [96,97]. Interestingly, the main mechanism of action by which dexamethasone prolongs nerve blockade and a clear dose response relationship both remain unclear. Different modalities, such as its anti-inflammatory properties, a vasoconstrictive effect or a modulation of signal transmission in the C-fibers, have been discussed [98]. Intravenously administered dexamethasone was found to provide similar effects compared to perineural administration [99]. It is worth mentioning that a recently published randomized double-blinded study found that the co-administration of dexamethasone for pectoral nerve block type II in patients undergoing a unilateral mastectomy did not influence postoperative opioid consumption over 72 h, nor did it prolong nerve blockade at all [100]. A lack of a longer-lasting analgesia was supported by two trials in volunteers, which compared the perineural effect of dexamethasone with its intravenous administration and with LA alone. Dexamethasone did not prolong the inhibition of nerve conduction when administered perineurally or intravenously [98,101].

Given the inconsistent results for dexamethasone’s off-label perineural use, its unclear mechanism of action, its potential to precipitate when combined to long-acting LAs, or its possible increase in LA-induced neurotoxicity, the systemic route should be preferred [84,85,98,102].

### 4.3. Epinephrine

Epinephrine is one of the oldest additives used to prolong PNB. The main mechanism of action is the α_1_-mediated vasoconstriction that decreases blood flow and thereby decreases the systemic reabsorption of the perineural LA [91]. A central α-mediated direct analgesic effect is also described [103]. However, the perineural administration of epinephrine alone has shown no analgesic effect [87]. The co-administration of epinephrine together with short-to medium-acting LAs such as lidocaine or mepivacaine increased the duration of action by more or less one hour [91]. No or only slight prolongation has been described for long-acting LAs [91]. Epinephrine decreases perineural blood-flow, raising the question of neurotoxicity in particular for patients at risk for nerve injury, such as diabetic patients [87]. Historically, epinephrine was added to LAs for the early detection of intravascular injection. With the widespread use of sonography employed for PNBs and the resulting reduced risk of deleterious intravascular injection, this indication has become less important.

Taken together, given the only short prolongation of action when combined with short- to medium-acting LAs, the use of epinephrine for PNBs is not recommended.

### 4.4. Opioids

The perineural use of opioids failed to demonstrate a solid effect above their systemic effects, and is therefore currently not recommended [72,87,104]. The only exception is buprenorphine, a lipophilic opioid with partial μ—opioid receptor agonist (MOP) and κ—opioid receptor agonist (KOP) activity. Apart from MOP-generated action on unmyelinated C-fibers, a concentration-related blockade of voltage-gated sodium channels has been discussed [84,105]. A recent meta-analysis described a prolongation of analgesia of nine hours when buprenorphine was added to LAs. This effect seemed to be more pronounced when buprenorphine was administered perineurally compared to its systemic application. Of note, the authors emphasized that these findings should be interpreted with caution due to the heterogeneity of the studies included [106]. Also, a five-time increased incidence of PONV was reported when buprenorphine was used [106]. Buprenorphine may in fact be a promising additive; however, solid evidence for its perineural use is still lacking.

## 5. Technical Measures to Prolong Analgesia

### 5.1. Continuous Techniques

The use of continuous techniques by means of perineural catheters has been described for all regional anesthetic techniques (epidural and spinal anesthesia, PNB). In modern perioperative care, epidural and peripheral catheters are used to prolong postoperative analgesia by the continuous infusion of low-concentrated LAs (e.g., ropivacaine 0.2%). Electronic delivery systems allow for additional boluses on demand (patient-controlled analgesia). Although recently developed nerve block techniques (e.g., fascial plane blocks) have been introduced into clinical practice as safe alternatives to epidural analgesia, the latter remains an effective alternative for postoperative pain management after major upper abdominal and thoracic surgery [107]. Nevertheless, careful patient selection based on a risk-benefit discussion is mandatory [108].

The epidural administration of LAs and opioids provides efficacious analgesia during labor [109]. Likewise, peripheral catheters have been frequently used to provide optimal analgesia for various indications [110]. In terms of toxicity, continuous infusions of LAs are regarded as safe, since common infusion rates do not result in toxic LA plasma concentrations [111].

Nevertheless, their use has decreased for several reasons. Multiple pharmacologic interventions as part of a multimodal analgesic regimen seem to be equally effective when combined with single-shot nerve blocks. Moreover, catheters have a high incidence of dislocation, and their insertion can be challenging [112]. Other technical problems and side effects are leakage, pump malfunction, undesired motor block and local infection. All continuous techniques have in common a requirement of additional nursing to check for their effectiveness and side effects. Thus, staff shortages have further limited the use of these techniques.

### 5.2. Continuous Wound Infusion

Continuous wound infusion (CWI) has been established as a safe and effective alternative to traditional regional anesthesia techniques. CWI is mediated through specially designed multi-hole catheters, which are inserted into the wound mostly at the end of surgery. The underlying mechanism of action is a blockade of afferent nociceptive fibers around the surgical wound. A recent meta-analysis showed a favorable analgesic profile without severe complications for different types of abdominal surgeries [113]. CWI has gained an important role in perioperative pain management in several types of surgeries as part of a multimodal regimen [114].

### 5.3. Infusion Systems

Many devices are marketed to deliver LAs via catheters. Electronic systems offer a variety of settings, whereas disposable elastomeric pumps provide continuous infusion and fixed bolus rates, depending on pump model (e.g., 2 mL every 15 min). In general, several modes of drug administration are being used: continuous infusion, programmed intermittent bolus (PIB) and bolus on demand, and a combination of all of those (e.g., continuous infusion and bolus on demand). The optimal result would be complete analgesia without any or less motor weakness. Since LAs are non-selective for Na_v_ channels, differential effects for sensory and motor blockades are difficult to achieve. Various infusion regimes have been studied. So far, a specific setting for the peripheral infusion of LAs cannot be recommended, since current data are extremely heterogeneous [115]. Nevertheless, the intermittent application of large boluses is preferred when extended spread is warranted [116]. In contrast, PIB has proven to be beneficial for labor analgesia, resulting in a higher patient satisfaction, less motor block and a lower incidence of instrumental vaginal delivery [117,118,119]. Thus, many institutions consider PIB as a standard mode for epidural analgesia during labor.

## 6. Conclusions

In conclusion, thus far, many promising and innovative pharmacological developments have occurred with the aim to extend the maximal duration for RA without the need for and pitfalls of catheter techniques. Over the last decade, local and regional anesthesia have gained popularity as parts of numerous multimodal analgesia protocols, with the aim to reduce perioperative pain, decrease LOS, and decrease opioid consumption and prescription patterns. However, many RA modalities fall short when their durations of action are compared to systemic modes of analgesia. In contrast to regional anesthesia, opioids can easily be re-dosed. Thus, substances or substance combinations that prolong block action, in combination with a lower side-effect profile, are highly desired. However, none of the newly developed substances have yet to reach satisfactory results. The practice of adding different substances to LAs in an effort to prolong block duration is frequently conducted, but the evidence for its efficacy is very heterogeneous, and no conclusion on the safety of this practice can be reached at this point. Recent data on the chemical compatibility of some substance mixtures, particularly regarding their aptitude for crystallizing after admixture, raise additional concerns. Efforts to introduce novel sensory-selective LA agents are underway and show promising early results, but none of these substances have reached routine clinical practice thus far.

## Figures and Tables

**Table 1 jcm-12-01541-t001:** Overview of approved Extended-release formulations.

Trade Name	LA/Analgetic	Polymer Class	Approved Application	Year of Approval
EXPAREL^®^	Liposolmal Bupivacaine	Liposome	Field block infiltration, brachial block, femoral block [22,23,24]	2020
ZYNRELEF^®^	Bupivacaine/Meloxicam	Biological polymer	Needle-free wound application [25,26,27]	2020
SABER^®^ Bupivacaine	Sucrose acetate isobutyrate extended-release bupivacaine	Biological polymer	Subacromial injection [28,29,30]	2021
XARACOLL^®^ INL-001	Bupivacaine	Biological polymer	Needle-free wound application for inguinal-hernia repair [31,32]	2021

**Table 2 jcm-12-01541-t002:** Comparison of mechanism of action, potential prolongation of sensory block and side-effects of perineurally administered LA-adjuvants.

Adjuvants	Mechanism of Action	Prolongation of Sensory Block	Side Effects
Clonidine	Inhibition of hyperpolarization-activated cation current (I_h_); vasoconstriction	2.8 to 3.3 h	Sedation, hypotension, bradycardia
Dexmedetomidine	Inhibition of hyperpolarization-activated cation current (I_h_);	Up to 5 h	Sedation, hypotension, bradycardia
Dexamethasone	None	Contradictory results	Increased LA neurotoxicity Precipitation with long-acting LA
Epinephrine	vasoconstriction	Up to 60 min	Increased LA neurotoxicity
Opioids	μ − opioid receptor agonist generated action on C-fibers	Up to 9 h (low evidence)	Nausea, vomiting, pruritus

## Data Availability

Not applicable.

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
