# Peer review of "Peripheral Regional Anesthesia Using Local Anesthetics: Old Wine in New Bottles?"

_jcm, 2023, doi:10.3390/jcm12041541_

Round 1
Reviewer 1 Report
Well written review on the current trends in local anesthetics and adjuvants.
Topic: It is time for an unbiased review of the available options for prolonging local anesthetic action to provide prolonged analgesia without the need for catheter-based techniques, which have the drawbacks as the authors have described in this manuscript.
Compared with other published material: This attempt clarifies existing controversies over the available additives and their efficacy.
Methodology: They have made a sincere attempt to justify their conclusions.
Conclusions: The conclusions are consistent with the evidence and arguments presented. As I have described, the arguments are well substantiated by available evidence.
References: The references are appropriate.
It surely is a well written review of current value to the readers.
Author Response
Reviewer #1:
- Comment:
Well written review on the current trends in local anesthetics and adjuvants.
Topic: It is time for an unbiased review of the available options for prolonging local anesthetic action to provide prolonged analgesia without the need for catheter-based techniques, which have the drawbacks as the authors have described in this manuscript.
Compared with other published material: This attempt clarifies existing controversies over the available additives and their efficacy.
Methodology: They have made a sincere attempt to justify their conclusions.
Conclusions: The conclusions are consistent with the evidence and arguments presented. As I have described, the arguments are well substantiated by available evidence.
References: The references are appropriate.
It surely is a well written review of current value to the readers.
Response:
Thank you for your nice comments.
Reviewer 2 Report
General comments
I would like to congratulate the authors on this excellent review paper.
It has always been the desire to prolong the effect of a nerve block while shortening the onset time, reducing toxicity, and avoiding motor block (aim for early mobilization). The authors summarize the latest developments in this field in a review article that is up to date.
Specific comments
In the international context, drug safety legislation will differ. In German-speaking countries, the mixing of two drugs is considered reconstitution (if provided for in the technical information) or manufacture (if not provided for by the manufacturer). In the latter case, this entails the assumption of legal responsibility and special obligations to inform the patient (off-label use). Mixing of medications is subject to notification to the drug authority. Perhaps the authors can clarify this for international readers (For me, the information was sufficient.).
A frequently used measure for shortening the onset time is mixing with sodium bicarbonate. Perhaps the authors could briefly discuss this.
Page 8 / Line 353
Consider heading "infusion systems" without bullet point (omit “2.4.2.”)
Page 8 / Lines 355-6
… whereas disposable elastomeric pumps provide continuous infusion and fixed bolus rates depending on pump model (eg 2ml every 15min).
Conclusions: The conclusions consistent with the evidence and arguments presented.
References: The references are appropriate.
Tables: The tables are clear and easy to read. Table 2 does not contain a complete list of all adjuvants investigated (e.g. magnesium, NMDA receptor antagonists, etc. are missing). The authors had the explicit wish to deal only with the most frequently used substances (see lines 233-235). In my opinion, this is sufficient.
Author Response
Reviewer #2:
General comments
I would like to congratulate the authors on this excellent review paper.
It has always been the desire to prolong the effect of a nerve block while shortening the onset time, reducing toxicity, and avoiding motor block (aim for early mobilization). The authors summarize the latest developments in this field in a review article that is up to date.
Response:
Thank you for your constructive and important comments. We have tried to incorporate your suggestions accordingly and they have helped to improve the manuscript.
Specific comments
- Comment 1:
In the international context, drug safety legislation will differ. In German-speaking countries, the mixing of two drugs is considered reconstitution (if provided for in the technical information) or manufacture (if not provided for by the manufacturer). In the latter case, this entails the assumption of legal responsibility and special obligations to inform the patient (off-label use). Mixing of medications is subject to notification to the drug authority. Perhaps the authors can clarify this for international readers (For me, the information was sufficient.).
Response:
This is an important point. Obviously, the reader should reflect on the national legislation, when he administers any drugs or combinations, even more when they are not used as approved by legal authorities.
When mixing substances is deemed necessary, national legislation must be respected and any necessary step (such as to inform the patients of the off-label use, etc.) should be done in someone’s personnel interest.
We now underlined this argument as follows:
“Also, it should be taken in mind that by using drugs for other applications than covered by the approval of medical regulatory authorities such as FDA or EMA, the national drug safety legislation must be respected. In some countries a special patient consent for off-label use is mandatory. “ Page: 6, Line: 265 - 268
- Comment 2:
A frequently used measure for shortening the onset time is mixing with sodium bicarbonate. Perhaps the authors could briefly discuss this.
Response:
Thank you for this annotation. Indeed, sodium bicarbonate is often used to shorten the time to onset of especially short and intermediate acting LA (eg. lidocaine) when used for peripheral and central regional anesthesia. In this review we aimed to describe strategies to prolong the effect of PNB and therefore ropivacaine and bupivacaine should be the most suitable LA’s. The combination of sodium bicarbonate with ropivacaine or bupivacaine is prone to precipitation due to the alkalization. Also, with the upcoming of ultrasound-guided PNB, a reduction of onset of sensory block was observed. Therefore, we did not discuss this aspect, as the shortening of onset was, as I already said, not the main topic of this review.
We now added as follows “Some additives are also used to shorten the time of onset of LA, such as sodium bicarbonate, which is often added to short or intermediate acting LA (eg. lidocaine). As this review focusses on the prolongation of sensory block in PNB, additives used to shorten the time of onset will not be will not further discussed.“ Page: 6, Line: 251 - 254
- Comment 3:
Page 8 / Line 353
Consider heading "infusion systems" without bullet point (omit “2.4.2.”)
Response:
Thank you for noting this. By editing the manuscript by the editorial office some changes occurred. We controlled and adopted the arrangement of the headings and subheadings throughout the manuscript as they were initially intended. We hope that by doing so we did not defy with the journals policies.
- Comment 4:
Page / Lines 355-6
… whereas disposable elastomeric pumps provide continuous infusion and fixed bolus rates depending on pump model (eg 2ml every 15min).
Response:
We added according to your suggestion.
Page : 9, Line : 402
Conclusions: The conclusions consistent with the evidence and arguments presented.
References: The references are appropriate.
- Comment 5:
Tables: The tables are clear and easy to read. Table 2 does not contain a complete list of all adjuvants investigated (e.g. magnesium, NMDA receptor antagonists, etc. are missing). The authors had the explicit wish to deal only with the most frequently used substances (see lines 233-235). In my opinion, this is sufficient.
Response:
Thank for pointing this out. As you noted we covered only what we deemed to be the most common substances and we excluded those, that are suspected to be neurotoxic (midazolam, magnesium) or that lead to unfavorable side effects such as hallucinations or nausea (ketamine, neostigmine).
To clearify, we now added the following sentence:
“… omit those with suspected neurotoxic effects (eg. midazolam, magnesium) or unwanted side effects such as nausea and hallucinations (eg. ketamine, neostigmine).86 (Table2)”
Page: 6, Line: 262 - 264